# A Reinforcement-Learning-Based Model for Resilient Load Balancing in Hyperledger Fabric

**Reem Alotaibi** [1,*] , **Madini Alassafi** [1] , **Md. Saiful Islam Bhuiyan** [2], **Rajan Saha Raju** [2] **and Md Sadek Ferdous** [3,4]

1   Faculty of Computing and Information Technology, King Abdulaziz University, Jeddah 21589, Saudi Arabia
2   Department of Computer Science and Engineering, Shahjalal University of Science and Technology, Sylhet 3114, Bangladesh
3   Department of Computer Science and Engineering, BRAC University, Dhaka 1212, Bangladesh
4   Imperial College Business School, Imperial College London, London SW7 2BX, UK
*   Correspondence: ralotibi@kau.edu.sa

**Abstract:** Blockchain with its numerous advantages is often considered a foundational technology with the potential to revolutionize a wide range of application domains, including enterprise applications. These enterprise applications must meet several important criteria, including scalability, performance, and privacy. Enterprise blockchain applications are frequently constructed on private blockchain platforms to satisfy these criteria. Hyperledger Fabric is one of the most popular platforms within this domain. In any privacy blockchain system, including Fabric, every organisation needs to utilise a peer node (or peer nodes) to connect to the blockchain platform. Due to the ever-increasing size of blockchain and the need to support a large user base, the monitoring and the management of different resources of such peer nodes can be crucial for a successful deployment of such blockchain platforms. Unfortunately, little attention has been paid to this issue. In this work, we propose the first-ever solution to this significant problem by proposing an intelligent control system based on reinforcement learning for distributing the resources of Hyperledger Fabric. We present the architecture, discuss the protocol flows, outline the data collection methods, analyse the results and consider the potential applications of the proposed approach.

**Keywords:** blockchain; private blockchain; hyperledger fabric; load balancing; privacy; IoT; machine learning; reinforcement learning

## 1. Introduction

With the ever-increasing popularity of blockchain technology, it is being utilised to disrupt a number of application domains, i.e., healthcare systems [1], financial transaction systems [2], educational sectors [3], and many more. Some of these applications are targeted towards enterprise applications to solve certain business problems. These enterprise applications must satisfy a number of requirements such as enterprise-grade scalability, performance, and privacy. In the current state of blockchain research, private blockchain systems are generally more favoured for such enterprise applications as these systems perform better with respect to the highlighted requirements.

To cater to the needs of business enterprises, there have been a few initiatives to introduce a number of private blockchain platforms. Hyperledger is one such open-source and global collaborative effort [4]. It facilities an umbrella initiative under which different private blockchain platforms, such as Hyperledger Fabric [5], Hyperledger Burrow [6], Hyperledger Sawtooth [7], Hyperledger Iroha [8], and so on have been developed. Among all these platforms, Hyperledger Fabric is one of the most matured private blockchain platforms; hence, it is widely used in different applications and researches.

Due to the append-only nature of blockchain, the size of blockchain keeps growing, consuming more resources as time goes by. In addition, an organisation usually needs

to deploy a peer node to connect to a private blockchain platform. This peer node acts as the gateway for the blockchain application. If a large number of users suddenly uses this peer node simultaneously, this could lead to a situation where the peer node would consume all its resources to serve the users. To solve these issues, there must be a load balancing mechanism which could efficiently handle the resource utilisation of the peer node. Unfortunately, these crucial issues are mostly overlooked while adopting private blockchain systems. In this article, we present a machine-learning-based load balancing model for Hyperledger Fabric, which aims to address the issues highlighted above.

Hyperledger Fabric utilises a container-based approach to deploy its blockchain network. There are a number of existing works which explore the mechanisms for load-balancing containerised applications using machine learning techniques, paving the way to implement a machine-learning-based load balancing approach for Hyperledger Fabric. For example, the authors in [9] incorporated three different approaches (Q learning, Dynamic Q learning, and Model-based) of RL (reinforcement learning) in scaling up/down of container-based applications. Authors in [10] presented a predictive mechanism for automatic scaling of microservice applications which utilised machine learning models such as Linear Regression, Random Forest, and Support Vector Regressor. A deep learning approach, Bi-LSTM (Bidirectional Long Short-term Memory), was utilised in [11] to forecast the HTTP workloads. In [12], the authors represented a survey considering the orchestration of containers on the basis of machine learning approaches. In [13], the authors illustrated the advantages and disadvantages of two different techniques of RL on cloud platforms. In [14], the authors designed a framework where resources are dynamically allocated for data centres using Docker containers.

Unfortunately, the focus of these existing works was on the load balancing of containerised applications. Understandably, these works did not consider if their approach could be adopted for a private blockchain system such as Hyplerdger Fabric. In addition, we have not found any research that has explored the possibility of extending the approach presented in these works for any private blockchain system. Hence, to the best of our knowledge, there is no existing work which has explored the possibility of adopting a load balancing approach using a machine learning model for a private blockchain system such as Hyperledger Fabric.

The core contributions of this article are as follows.

- We present the first-ever proposal for a machine learning model for load balancing the resources of a leading private blockchain platform, Hyperledger Fabric.
- We discuss the architecture of the proposed method and analyse its protocol flow with data models.
- We outline the data collection methods along with a discussion of how an optimal model has been trained.
- Finally, we analyse the advantages of the proposed approach and envision how the proposed methods can be integrated within the Hyperledger Fabric blockchain platform for deploying an optimal load balancing mechanism.

The article is structured as follows. In Section 2, a brief introduction of blockchain, Hyperledger projects and Hyperledger Fabric is presented. The proposed approach along with its motivation, system architecture, data model and protocol flow are discussed in Section 3. Section 4 focuses on the conducted experiments and the analysis of the result. In Section 5, the advantages of the proposed model are analysed along with a discussion of how the proposed approach can be deployed within Hyperledger Fabric and possible future works. Finally, we conclude in Section 6.

## 2. Background

In this section, we present a brief background on blockchain (Section 2.1) and the Hyperledger Project (Section 2.2) along with a detailed discussion on different aspects of Hyperledger Fabric (Sections 2.3–2.6).

## 2.1. Blockchain

Bitcoin, the first successful decentralised digital currency in the world, introduced the notion of *blockchain* [15]. A blockchain is an example of a ledger which is distributed across a number of Peer-to-Peer (P2P) nodes [16]. This ledger consists of sequential blocks which are chained together using cryptographic mechanisms, thus representing an ordered data structure. Each block within a blockchain contains a number of transactions and a set of metadata, denoted as the *block header*. Within the block header, a cryptographic hash refers back to the previous block in the chain, thus forming a chain of blocks. To ensure that a blockchain is synchronised across multiple P2P nodes, a consensus algorithm is utilised. There are a number of distributed consensus algorithms such as Proof of Work (PoW), Proof of Stake (PoS), Practical Byzantine Fault Tolerance (PBFT), Proof of Burn (PoB), Delegated Proof of Stake (DPoS), and so on, each with its own advantages and disadvantages [17]. The consensus algorithm in a blockchain system also ensures the data immutability feature.

The next evolution of the Bitcoin blockchain is a new breed of blockchain system which supports the notion of *smart-contract* by integrating a computing platform (e.g., a virtual machine) with a blockchain [18]. A smart-contract is a computer program which can be deployed and executed using the corresponding computing platform of the blockchain system. As such, smart-contracts are tied to a blockchain system; they can be invoked autonomously using transactions. Consequently, the executions and generated results of such smart-contracts can become immutable and irreversible, which is regarded as sought-after properties in many application domains. Interestingly, such smart-contract supporting blockchain systems also offer some additional advantages: data persistence, data provenance, distributed data control, accountability, and transparency. Examples of these systems are: Ethereum [19], Cardano [20], Polkadot [21], and so on.

There are mainly two types of blockchains:

- Public blockchain: A public blockchain, also known as the *permissionless blockchain*, facilitates the mechanism by which anyone can join the network. Any user of this blockchain can submit transactions when they wish and participate in the block creation process. Examples of public blockchain systems are Bitcoin [22], Ethereum [19], Cardano [20], Polkadot [21], Litecoin [23], Monero [24], and so on.
- Private blockchain: A private blockchain, also known as *permissioned* blockchain, on the other hand, allows only authorised, identified and trusted entities to participate in different activities within the system. These users generally have different types of permissions, and blockchain establishes access control rules for each user. The ultimate goal of such blockchain systems is to ensure the privacy of different transactions and provide better performance and scalability in comparison to any public blockchain systems. Examples of private blockchain systems are Hyperledger Platforms [4], Quoram [25], and others.

## 2.2. Private Blockchain System: Hyperledger

Hyperledger is an open-source and global collaborative project [4] hosted by the Linux Foundation. It consists of different leading partners from the Internet of Things, banking, finance, supply chains, manufacturing, and technology platforms. The Hyperledger initiative provides an umbrella mechanism under which different industry partners can collaborate to develop different types of efficient, reliable, scalable, and high-performance private blockchain platforms. In addition, the Hyperledger initiative collaborates with different industry partners to establish standards and guidelines for adopting blockchain systems within different business processes. Under this umbrella initiative, there are a number of various projects, such as Hyperledger Fabric [5], Hyperledger Burrow [6], Hyperledger Sawtooth [7], Hyperledger Iroha [8], and so on. Among all these platforms, Hyperledger Fabric is one of the most matured ones which is widely used in different applications and research. That is why we have focused on Hyperledger Fabric for this research. However, it is to be noted that the approach presented here would be suitable for

any other private blockchain platform after minor modifications. Next, we discuss different aspects of Hyperledger Fabric.

### 2.3. Hyperledger Fabric (HF)

Hyperledger Fabric (HF) is a general-purpose private blockchain platform suitable for a number of enterprise-level use-cases. One major strength of HF is that it supports a number of general purpose programming languages, such as Java, JavaScript, Go Lang and others, for writing smart-contracts (known as chaincode in HF terminology). To ensure the privacy among different organisations even within the same blockchain network, HF utilises a unique concept called *channel*. A channel in HF allows users to create and manage multiple blockchains as required by different organisations within the same network, and thus privacy can be ensured. This can be a crucial property in many application domains where activities between different organisations must remain private. In addition, the consensus algorithm is modular and pluggable in HF, meaning different consensus algorithms can be integrated as required by the application. Some of the consensus algorithms that HF currently supports are SOLO and Kafka [26] with Simplified Byzantine Fault Tolerance (SBFT); [27] is to be added soon.

### 2.4. Core Components of HF

Hyperledger Fabric consists of a number of components. In the following, we briefly present the functionalities of its different components.

#### 2.4.1. Nodes

There are different types of nodes in Hyperledger Fabric as discussed below:

- Certificate Authority (CA): A CA is responsible for offering an identity service, called *Membership Services Provider (MSP)*, to identify each entity within the network. All other nodes and users must be registered with the MSP of the corresponding CA before they can interact with a Fabric platform. Once registered, the public–private key pair and the cryptographically validated digital certificate for each entity (a node or a user) are generated and distributed [28]. Then, the entity needs to use these to interact with other Fabric components.
- Peer: A blockchain network is constructed with a set of peer nodes where each peer is responsible to receive a block from an orderer (discussed later) and after validating, adds the block to the blockchain. Thus, each peer holds a copy of the full blockchain and provides deliberate redundancy to the blockchain system.
- Endorser: An endorsing peer (or endorser in short) is responsible for validating each transaction. Each endorser utilises a policy (see below) to check if a certain user is authorised to submit a transaction.
- Orderer: An orderer collects different transactions from the network and combines them into a block [29]. Then, it sends the block to all the peers belonging to a certain ledger.

#### 2.4.2. Chaincode, Ledger, and Channel

A smart-contract is known as a chaincode in HF terminology [30]. As mentioned earlier, such a smart-contract is a self-executing autonomous program. It essentially encodes the rules of specific types of business functionalities as required for the blockchain platform and its associated application to function .

The ledger represents the blockchain within the Fabric network [31]. It is a sequenced, tamper-resistant record of all transactions occurring in the network where different transactions are structured within a block.

Fabric utilises a unique concept called *channel* [32] to add a layer of privacy. A channel creates a subnet within the network and initiates a separate ledger. In this way, different ledgers can be maintained within the same blockchain network. Each peer is essentially attached to at least one ledger via one channel. However, a peer can be part of multiple

ledgers with the help of multiple channels. Fabric provides the mechanisms by means of a policy (discussed later) which dictates who or which organisation can access which ledger.

### 2.4.3. Policy

A policy in HF dictates how a particular Fabric blockchain is governed and which entities within the network have what type of capabilities [33]. For example, policies are utilised to decide which organisation within the network has access to which resources. In addition, Fabric uses policies to facilitate an access control mechanism for each user.

### 2.4.4. DApp

A DApp (Decentralised Application) is an essential component in every blockchain application. A DApp serves two purposes: (i) on the one hand, it is attached to a peer of the blockchain network so that it can interact with the blockchain, and (ii) on the other hand, it exposes web APIs (Application Programmer Interfaces) so that other web or mobile applications can interact with the blockchain via the API.

### 2.5. Interactions among HF components

Next, we briefly present how different Fabric components interact with each other when a transaction is submitted. Figure 1 summarises the flows of activities as discussed next. To submit a transaction, a user (of an organisation belonging to the Fabric network) utilises a peer. Once submitted, the peer forwards the transaction to the endorser(s) (Illustrated in Figure 1 as steps 1, 2, and 4). An endorser validates the transaction by checking if the user is allowed to perform the requested action in the ledger as encoded within the transaction (steps 3 and 5 in Figure 1). Once validated, the peer forwards the validated transaction to the orderer(s). The orderer checks the validated transaction and creates a block with the transaction(s). Then, the orderer sends the block to the endorsers and peers. Each peer and endorser adds the block to the blockchain, and this updates the state of the ledger (steps 6 and 7 in Figure 1). Finally, the user receives a response.

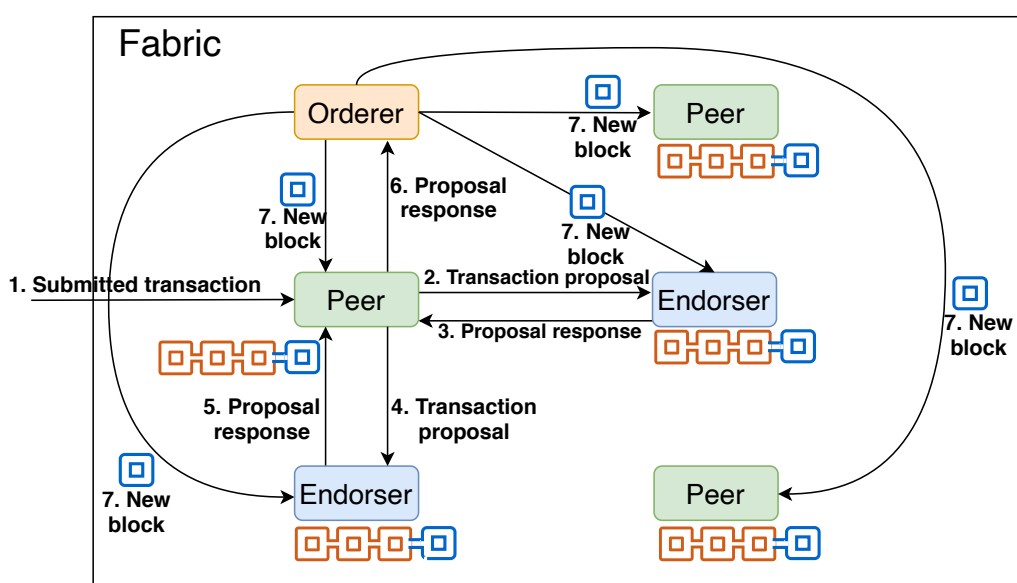

**Figure 1.** Flow of activities in Fabric

### 2.6. Deployment Approach in HF

An example of HF deployment in an enterprise use-case between multiple organisations is illustrated in Figure 2. As per the figure, there are four organisations. Each organisation will need to utilise a peer to connect to the blockchain network and a DApp to interact with the blockchain via the peer. In most cases, the DApp will be hosted within the peer so that the organisation does not need to maintain two nodes. A user then either

uses a UI (a web browser or a mobile app) to interact with the blockchain application via the DApp. Here, the blockchain application can be thought of as a combination of DApps and one or more chaincode within the blockchain platform. The nodes within the blockchain network is usually deployed using a cluster with Kubernetes [34] or Docker container swarms.

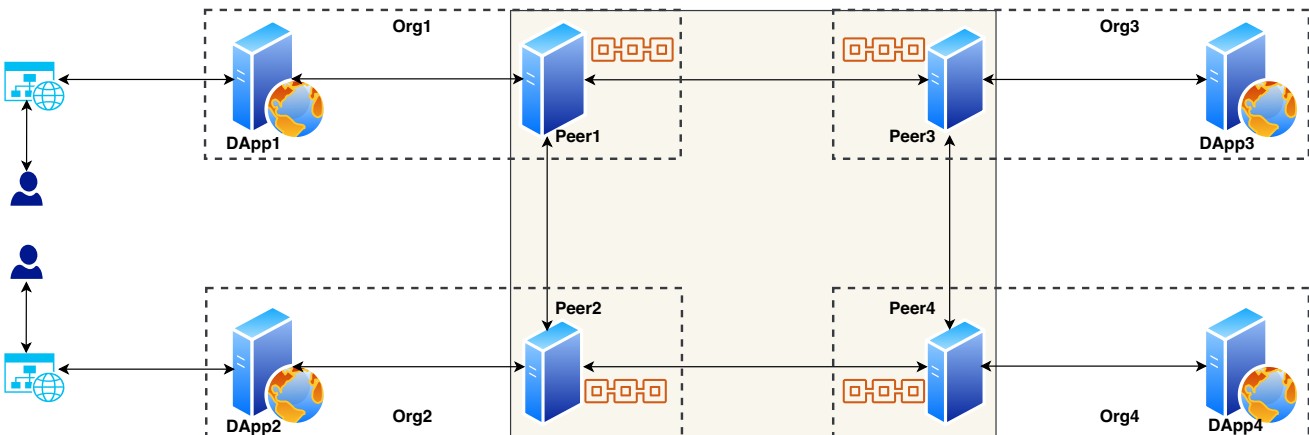

**Figure 2.** Fabric deployment.

## 3. Proposed Approach

In this section, we present the motivation (Section 3.1) of our work, the system architecture and implementation details (Section 3.2) and protocol flow (Section 3.3).

### 3.1. Motivation

As highlighted earlier, there is one important issue in every blockchain technology: due to the nature of immutability and the append-only data structure, the ledger inevitably grows as time goes by. In addition to that, the consumption of other computing resources, such as CPU, memory, and storage, must be monitored. This is particularly true for private blockchain systems where a DApp and a peer might need to serve a large user-base (Figure 2). That is why it is important to monitor the resource consumption of different nodes within a private blockchain platform, particularly for peer and orderer nodes.

A major sign of exhausted resources within a peer is if the peer struggles to keep up with the incoming transaction load while performing relatively simple tasks, i.e, querying the ledger [35]. Similarly, as throughput increases on an ordering node (or orderer), its resources can become exhausted. This is because all blocks in Fabric are ordered by the orderer. If it is observed that an orderer struggles with throughput, it might be a sign that resources might need to increase for its availability [36].

Monitoring resource consumption on these nodes is not enough. There should be a mechanism to act when the monitoring detects a resource being exhausted on a particular node. For example, if the storage for a peer is exhausted, there must be way to deploy a new peer with a larger storage allocation and let the ledger sync. Our research presented in this article essentially tackles this particular issue. It presents a novel reinforcement learning machine learning which could be utilised to create a resilient load balancing mechanism for Hyperledger Fabric. Thus, once this model is integrated, nodes within the Fabric network could be automatically scaled up and down promptly based on their current resource consumption parameters.

To scale a container-based application, there are popular approaches such as threshold-based rules [37], control theory [38], time series analysis [39], and reinforcement learning [40]. However, in this work, we present and implement a reinforcement-learning-based method for Hyperledger Fabric so that its scalability issue can be minimised and resource consumption can be optimised.

### *3.2. System Architecture and Deployment*

The top-level architecture for the proposed approach is illustrated in Figure 3. It has four separate components: blockchain network, DApp, Job Scheduler, and ML module. The functionalities of each component and how these components have been deployed are discussed below.

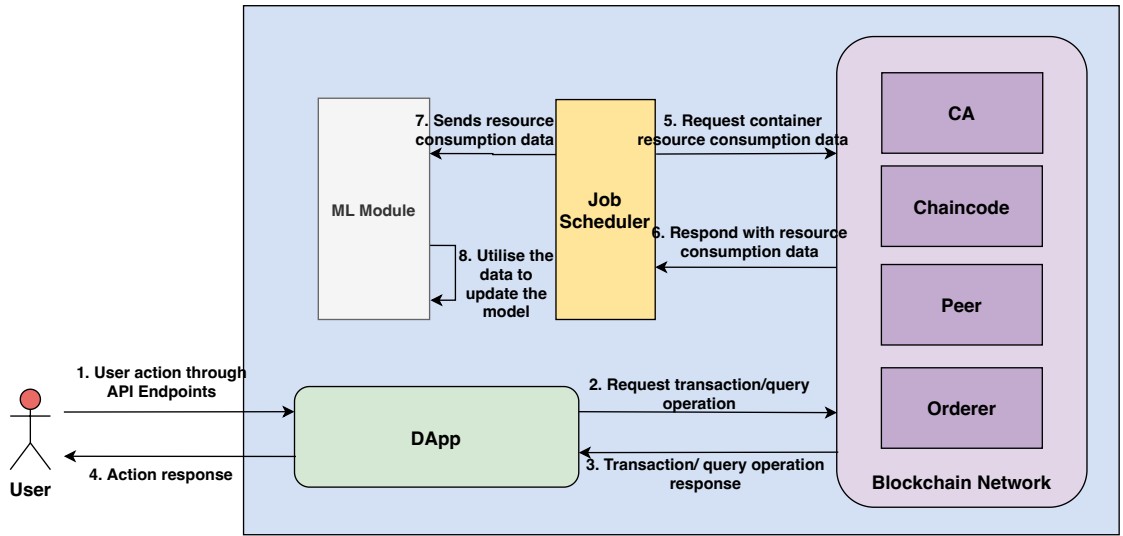

**Figure 3.** System architecture.

### 3.2.1. Blockchain Network

The blockchain network in the architecture represents an HF blockchain network consisting of a number of nodes, such as CAs, Peers (including endorsers), and orderers. They carry out the functionalities as discussed earlier.

During the implementation, we created a consortium blockchain network consisting of two organisations, named *Org1* and *Org2*. These two organisations represent a car supply chain where *Org1* plays the role of a car manufacturer, whereas *Org2* is a car distributor. Org1 can add a car when manufactured, and Org2 can query all the available cars ready to be sold.

Within the implemented network, there is a CA, an orderer, and a peer for each organisation. Each peer also acts as the endorsing peer. In addition, there are also chaincode and CouchDB containers for each organisation. All peers are connected via a single channel. The current version of Hyperledger Fabric supports a RAFT-based CFT (Crush Fault Tolerant) consensus algorithm. We used this algorithm in our experiments.

### 3.2.2. DApp

Each peer within the organisation is connected to a DApp. It has two sub-components: (i) API component exposes API interfaces for the applications, and (ii) the blockchain component interacts with the blockchain platform. With these two components, DApp actually facilitates the *Business Service* which encapsulates the core business functionalities of the blockchain application, e.g., car creation and car query. It exposes three API end points for any application and interacts with the chaincode to execute the functionalities required for these end points. These end points are:

- createCar: Generates a transaction when a new car is added to the organisation.
- queryCar: Queries all the available cars to be sold.
- createUser: Creates a user for a particular organisation.

The DApp has been developed with the Fabric NodeJS SDK.

### 3.2.3. Job Scheduler

The Job Scheduler serves two purposes:

- To collect resource consumption data (e.g., CPU, memory and, storage) for each container.
- To feed these data into the machine learning module (discussed later).

The Job Scheduler was also developed with the Fabric NodeJS SDK.

### 3.2.4. ML module

The ML component is deployed as a separate ML application which interacts with the Job Scheduler to receive resource consumption data and prepare the ML model. In traditional supervised or unsupervised machine learning approaches, a set of static data is selected firstly. Then, a model is built using the static data set. This type of model can only learn what the data set contains, and hence, it cannot capture any new scenarios that might arise and adjust accordingly. Here, we incorporate a special algorithm named Contextual Multi-Armed Bandits (Algorithm 1) [41] under reinforcement learning (RL) [42]. It is an iterative learning approach. In every iteration, an agent receives a context or feature vector, performs an action, observes a reward, and updates internal parameters. An RL-based agent does not require a pre-annotated data set like any supervised learning method. Instead, it expects rewards from the environment. This is a better approach in comparison to any data annotation approach, as data annotation might need manual hand engineering which can also add biases in the data set . In our system, at every 10 s, the job scheduler module interacts with each container to collect resource data for the respective container and sends the data back to the ML module.

The contextual bandit algorithm performs as follows. At time $t = 1, \ldots, n$ a new observation or context $X_t \in \mathbb{R}^d$ comes from the environment and is given to the agent. The agent suggests one of the k possible actions, $a_t$ based on its internal model and $X_t$. After that, a reward $r_t$ is generated based on $a_t$ and given back to the agent. The agent receives the reward as feedback and updates its internal model according to the reward. The agent always tries to increase the mean reward. The pseudocode of the contextual bandit algorithm is provided below:

---

**Algorithm 1** Multi-armed contextual bandits

---

1: **begin**
2: 　　**for** $t \leftarrow 0$ **to** $\infty$ **do**
3: 　　　　*Observe context $X_t \in \mathbb{R}^d$*
4: 　　　　*Retrieve model $\theta_t$*
5: 　　　　*Action $a_t \leftarrow BestAction(X_t, \theta_t)$*
6: 　　　　*Compute reward $r_t$ based on $a_t$*
7: 　　　　*Renovate model with $(X_t, \theta_t, r_t)$*
8: 　　**end for**
9: **end**

---

### 3.3. Protocol Flow

Now, we present the protocol flows which illustrate how users and different components of the proposed system interact with each other. However, first we introduce mathematical notations (presented in Table 1) and the data model (presented in Table 2).

Data Model: The proposed system represents a request–response model where a response is generated in accordance with a request. A request in the system is denoted with *req* in Table 2, and it consists of two components: *type* and *data*. Here, *TYPE* denotes different request types and *type* $\in$ *TYPE*. On the other hand, *DATA* represents different data corresponding to each request type and *data* $\in$ *DATA*. The definitions of *TYPE* and *DATA* are presented in Table 2.

**Table 1.** Cryptographic notations.

| Notations | Description |
| --- | --- |
| $K_U$ | Public key of the user. |
| $K_U^{-1}$ | Private key of user. |
| $N_i$ | A fresh nonce. |
| $\{\}_K$ | Encryption operation using a public key $K$. |
| $\{\}_{K^{-1}}$ | Signature using a private key $K^{-1}$. |
| $[]_{https}$ | Communication over HTTPS channel. |

**Table 2.** Data model.

| |
| --- |
| $req \triangleq \langle type, data \rangle$ |
| $TYPE \triangleq \langle createCar, queryCar \rangle$ |
| $DATA \triangleq \langle createCarData, queryCarData \rangle$ |
| $createCarData \triangleq \langle carNumber, company, model, colour \rangle$ |
| $queryCarData \triangleq \langle carNumber \rangle$ |
| $resp \triangleq \langle \text{A message} \vee queryCarData \rangle$ |
| $stats \triangleq \langle CPUusage, Memoryusage \rangle$ |

Next, *createCar* in type implies that it is a request for adding a new car with the data set denoted with *createCarData*. In *createCarData*, (*carNumber*) represents the unique number of a car which can be used to identify one single car, (*company*) refers to the manufacturing company name, (*model*) refers to the car model name, and (*colour*) refers to the colour of the car. This implies that a *createCar* request must contain a *number*, *company*, *model*, *colour*, and *owner*. Similarly, *queryCar* in type implies that this request will return the queried car with *queryCarData*. In *queryCarData*, (*carNumber*) represents the unique number of the car that is used to add the new car. Finally, *resp* represents a response which may contain a message or *createCarData*. On the other hand, *stats* represents the (*CPUusage*) and (*Memoryusage*) data.

It is to be highlighted that the *req*, *resp*, and its respective contents utilised in the flows correspond to business services, and *stats* utilised in flows relate to the interactions involving the job scheduler and the ML module.

Algorithms: Next, we present Algorithm 2 which encapsulates the functionalities of the chaincode for facilitating the business services between the car manufacturer and the car distributor.

When a request (*req*) is received by the car chaincode (denoted with *Car CC* in Algorithm 2), it initiates its invoke function (line 2 in Algorithm 2). Within this invoke function, *data* and *type* from *req* are retrieved, and any of the two functions, *carQFunc*, and *createCarFunc* are invoked depending on the request type (line 5 to 9 in Algorithm 2). For example, the *carQFunc* encodes the logic for querying about a car, whereas the *createCarFunc* encodes the functionality of adding a new car. If no data type is matched, then null is returned (line 10). Once completed, the *Car CC* algorithm returns a response (*resp*) back to the DApp (line 12).

Protocol Flow: Now, we present the protocol flow involving different components of the proposed architecture. The business service encodes the interactions of a typical blockchain application, in our case a car management application. As different users use this application, the job scheduler and the ML module work in the background to collect resource consumption data and feed the data to the ML module to train the ML model.

---

**Algorithm 2** Car CC: / / ▷ Chaincode

---

    **Input:** *req* → the request from the user
    **Output:** *resp* → the chaincode generated response

1: **begin**
2:     **function** invoke(*req*)
3:         *data := req.data*
4:         *type := req.type*
5:         **if** req.type == queryCar **then**
6:             *resp* = carQFunc(*data*);
7:         **else if** req.type == createCar **then**
8:             *resp* = createCarFunc(*data*);
9:         **else**
10:            *resp* = (*null*);
11:         **end if**
12:         send *resp* back to user;
13:     **end function**
14:     **function** carQFunc(*data*)
15:         **if** data.carNumber **then**
16:            *carInfo* = *getState*(*data.carNumber*);
17:            **return** *carInfo*;
18:         **end if**
19:         **return** 'Car number not given';
20:     **end function**
21:     **function** createCarFunc(*data*)
22:         *putState*(*data.carNumber, data*);
23:         **return** 'New Car Created';
24:     **end function**
25: **end**

---

Within the business service flow, to interact with the blockchain network for submitting a transaction, e.g., for adding a car entry, every user must have a private key. Generally, every user is registered at the HF network at first, and then the MSP provides the key pair for the user. We have skipped this protocol flow for brevity and mostly focused on the core functionality of adding a car and querying a car. The protocol flow of creating a car is illustrated in Figure 4. As per the flow, a user signs *req* (represents either a request for car creation or car query) with $K_{U_f}^{-1}$ and transmits it to DApp over an HTTPS channel. DApp interacts with the MSP to retrieve the public key of the user ($K_U$) which is then used to validate the signature. DApp then sends *req* to *CCC*. Depending on the *req* type, either *carQFunc* or the *createCarFunc* with (line 5 and 8 in Algorithm 2) is invoked. With the *carQFunc* function, the car data of the corresponding car are retrieved from the blockchain (line 14-20 in Algorithm 2). With the *createCarFunc*, new car data are stored in the blockchain (line 21-24 in Algorithm 2). A successful car creation operation will return a "*New car created*" response which will be returned back to the user over an HTTPS channel.

Next, we present the protocol flow involving the interactions between the job scheduler, blockchain Docker containers, and the ML module. The protocol flow for these interactions is illustrated in Figure 5. At a predefined interval, the job scheduler sends a query for stats to the containers. The containers return the *stats* with the CPU usage and memory usage data in percentage. Then, the job scheduler sends this data (*stats*) to the ML module. The ML module utilises the data to prepare a suitable ML model. The job scheduler feeds the data to the ML module to train it until it shows an optimal result. At this point, it is considered that an ML model has been found.

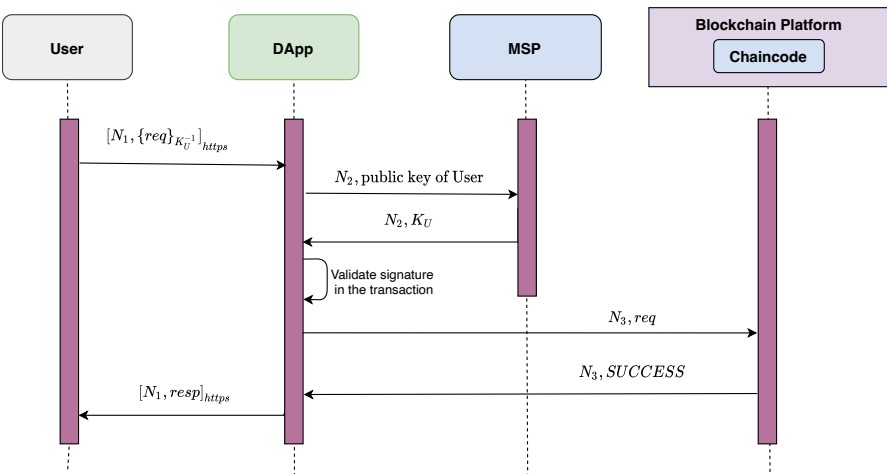

**Figure 4.** Car creation and query flow.

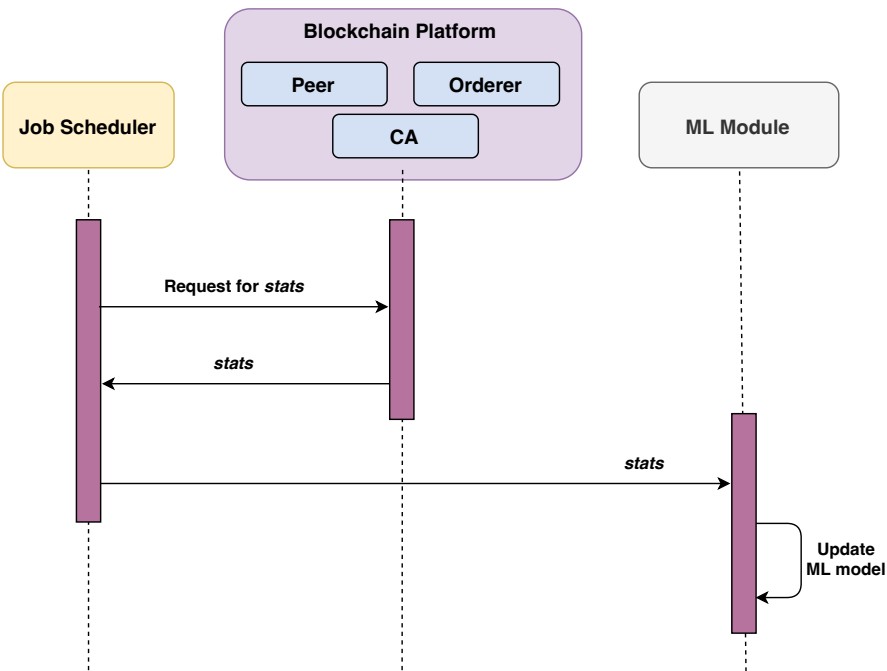

**Figure 5.** Blockchain network with ML model.

## 4. Experiment and Performance Analysis

To carry out the experiments involving our proposed architecture, it is important to find the optimal configurations under which the experiments can be carried out. For this purpose, we have utilised Hyperledger Caliper [43]. Caliper is a blockchain benchmarking tool for Hyperledger-based blockchain platforms, including Hyperledger Fabric. Using Caliper, the performance of any blockchain implementation can be measured under a predefined network configuration.

After experimenting and testing with different Caliper configurations, we have identified and opted for the following optimal configurations. The *Batch Timeout* (signifying the amount of time to wait before creating a batch of transactions) is set as 2 s with the maximum message count for a batch set to 500. The transaction rate is set to 60 to 100 per second. In each iteration with five users, the transaction rate is increased by 10. Under these configurations, we carried out each experiment four times. The results of these experiments are then averaged. The generated result is presented in Figure 6 where the X axis represents the transaction request rate sent by users, and the Y axis shows the actual transaction per second (TPS).

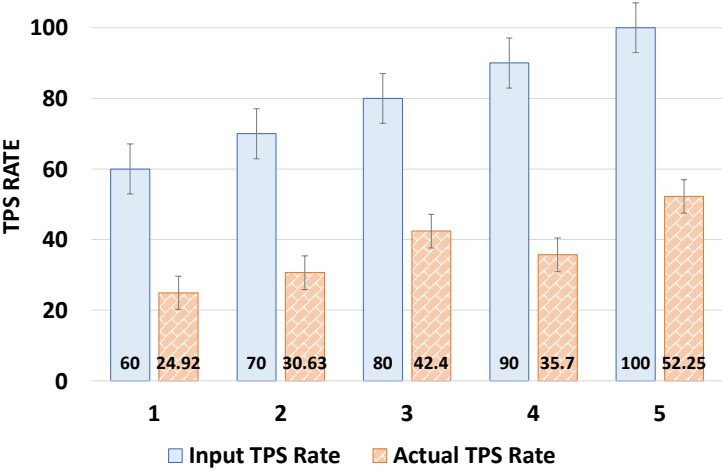

**Figure 6.** Performance on TPS.

Once the optimal configurations were fixed, the next step was to carry out the main experiments involving the proposed architecture. For these experiments, to deploy the HF network, a Docker swarm cluster was deployed on a desktop having the following configurations:

- Ubuntu 18.04-64 OS,
- Intel(R) Core i5-8265U @1.60 GHz quad-core CPU,
- 8 GB DDR4 RAM,
- 256 GB SSD,
- 1 TB HDD,
- 2 GB GeForce MX150 Graphics GPU.

The resources allocated for each network entity under the Docker swarm cluster node are shown in Table 3. As mentioned earlier, we simulated a scenario where five users from two organisations submitted different transactions via the DApp for creating cars and querying cars, essentially simulating the business service logic. While these transactions were being submitted, the job scheduler module would collect resource consumption data from different entities and the steps discussed above were followed to optimise the ML model.

**Table 3.** Blockchain network resource allocation

| Container Name | CPU Limit (in CPU Share) | CPU Reservation | Memory Limit (in MBs) | Memory Reservation |
|---|---|---|---|---|
| Peer 1 Org1 | 0.10 | 0.05 | 100 M | 50 M |
| CouchDB 1 | 0.20 | 0.05 | 150 M | 50 M |
| Peer 1 Org2 | 0.10 | 0.05 | 100 M | 50 M |
| CouchDB 2 | 0.20 | 0.05 | 150 M | 50 M |
| orderer 1 | 0.10 | 0.05 | 100 M | 50 M |
| CA Org1 | 0.10 | 0.05 | 100 M | 50 M |
| CA Org2 | 0.10 | 0.05 | 100 M | 50 M |

### 4.1. Data Collection

For the data collection phase, simulated scenarios were created where users from two organisations either created new cars by utilising the *createCar* end point or submitted queries using the *queryCar* end point. Among all the requests, 70% were for creating cars and 30% were for querying cars. This is because creating a car would require submitting a transaction thereby consuming more resources, while querying a car does not require submitting any transaction. Different scenarios were simulated for a period of around 7 h. The requests varied from time to time during this period. For one time period, a higher number of requests were generated, and for another period the number of requests was reduced. In addition, no requests were sent to the end points for some periods. Thus,

all possible situations were covered while data were being collected. Figure 7 shows the collected data for the container resources during the simulated experiment. The blue line in Figure 7 represents the memory usage, and the green line represents the CPU usage in percentage. In this whole data collection phase, a total of 2651 transactions were successfully submitted, and 1007 blocks were created for these transactions.

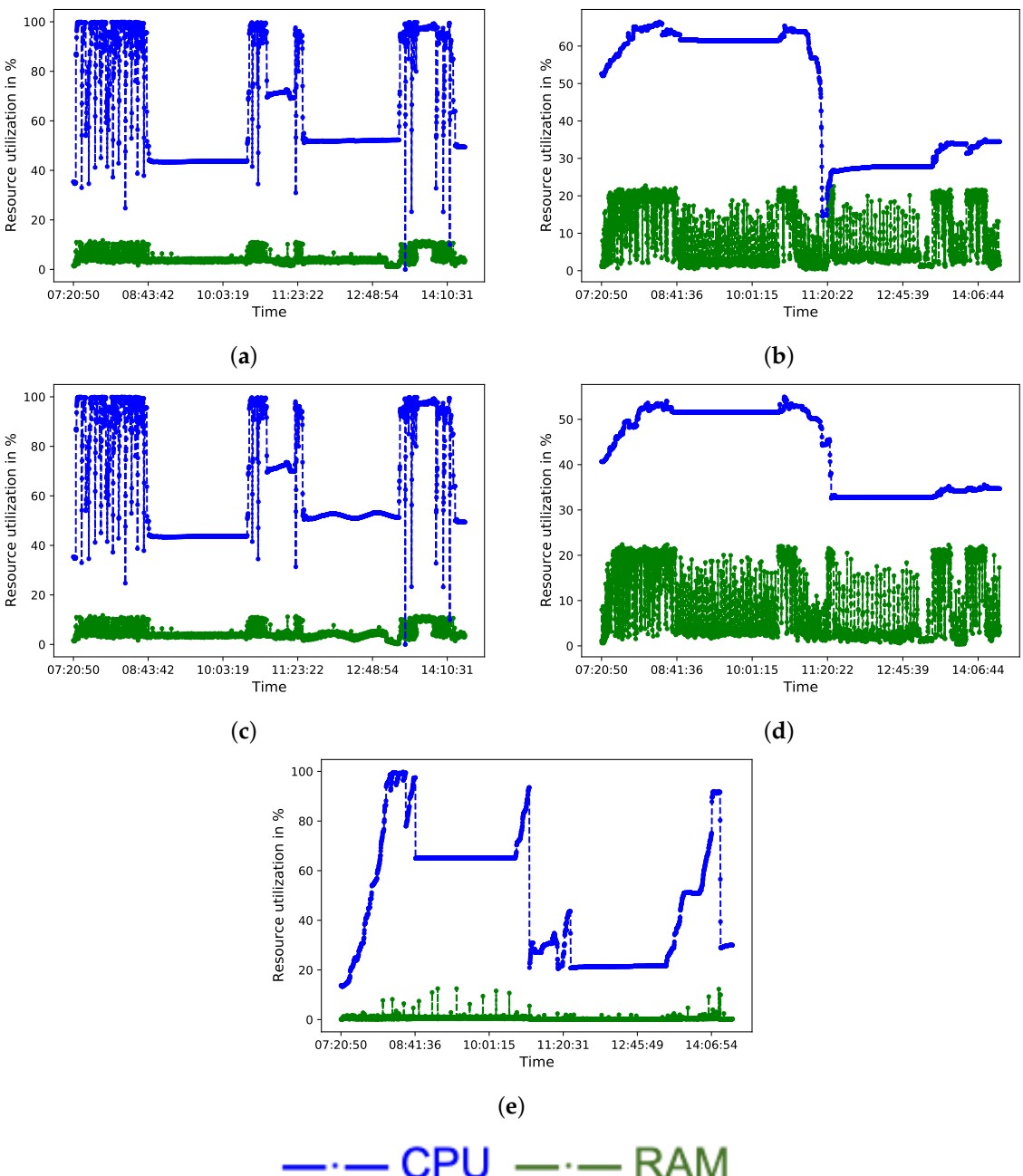

**Figure 7.** RAM and CPU utilization in percentage scale with respect to time: (**a**) Peer 1 Org 1; (**b**) CouchDB for Peer 1 Org 1; (**c**) Peer 1 Org 2; (**d**) D CouchDB for Peer 1 Org 2; (**e**) Orderer.

Figure 7a shows the container resources of Peer 1 of organisation 1 (Org 1 in short) and the corresponding world-state database container data are shown in Figure 7b. As can be seen from the beginning in Figure 7a, Peer 1 of Org 1 receives a high of number requests resulting in spikes in memory and CPU usage. When the memory consumption was increased to 100% of its capacity, the peers container crashed due to the memory out

of bound issue. The container started after a few seconds and started receiving requests again. From Figure 7b, we can see the resource consumption for the world-state CouchDB (the ledger) associated with Peer 1 Org 1, and it consumes its memory gradually with high throughput as the time goes by. After a period at 08:43, we stopped sending requests to peers, and the resource consumption stayed the same. After 10:30, we started sending requests again, however, with a mixed send rate.

The graphs shown in Figure 7c,d represent the resource consumption of Peer 1 of Org 2 and the CouchDB container associated with it, respectively. As these containers received requests similar to Peer 1 of Org 1, these graphs are also similar to the graphs of Peer 1 of Org 1. In Figure 7e, the resource consumption of the orderer is shown. As the orderer receives more requests, the resource consumption increases gradually until the requests to the orderer stopped coming.

### 4.2. Feature Extraction

The data set contains the percentage usage of RAM and CPU at every 10 s interval for different containers. However, we do not only consider a single time moment record, rather we inspect a fixed window of previous time moment records for understanding the current resource usage trend. If the current RAM usage is $R_t$, then we firstly take six records $(R_{t-5}, R_{t-4}, \cdots, R_t)$ including the current one as our predefined window length is six. Then, we generate five features taking deviations with respect to $R_{t-5}$. The first feature value is $R_{t-5} - R_{t-4}$ which denotes the increment in RAM usage compared to 40 s ago. The second feature value is $R_{t-5} - R_{t-3}$ and so on. A similar strategy is applied for the CPU usage as well. If the current CPU usage is $C_t$, then we extract these records $C_{t-5} - C_{t-4}, C_{t-5} - C_{t-3}, \cdots, C_{t-5} - C_{t-1}$. Thus, these 10 features extracted from the data set are then used with the selected reinforcement learning model.

### 4.3. Result

We adopt an average reward to evaluate the performance of the agent. After each iteration, the agent receives a reward based on how intelligently it chose the action at that particular iteration. For example, when CPU and RAM usage are high and the agent takes the action to scale down containers, then it will receive a reward close to 0.0. Similarly, when the agent takes the scale up action instead in the same situation, it will receive a positive reward close to 1.0. Consequently, an average reward after $t$ iteration is defined by

$$\bar{R}_t = \frac{\sum_{i=1}^{t} R_i}{t}; \ \bar{R}_t \in [0.0, 1.0]$$

As the agent continuously receives rewards and fine-tunes its understandings, it is expected that the average reward trend will be increasing with the increment of iteration, and it is always expected that average reward will converge to one. We have trained six agents separately. Figure 8a–e represent the average reward curves for Figure 7a–e, respectively.

The one common thing is that each agent's mean reward trend is increasing and converging to 1.0 after a certain number of iterations, and before that we can observe many fluctuations. The reason for fluctuations at the starting is that the agent did not experience sufficient scenarios to perform properly. From the example in Figure 8, we can see that the mean reward in Figure 8a,e starts to increase smoothly after approximately 100 iterations; however, it takes less steps in others. The reason behind this phenomenon is because of the many fluctuations in percentage usage of RAM for Figure 8a,e.

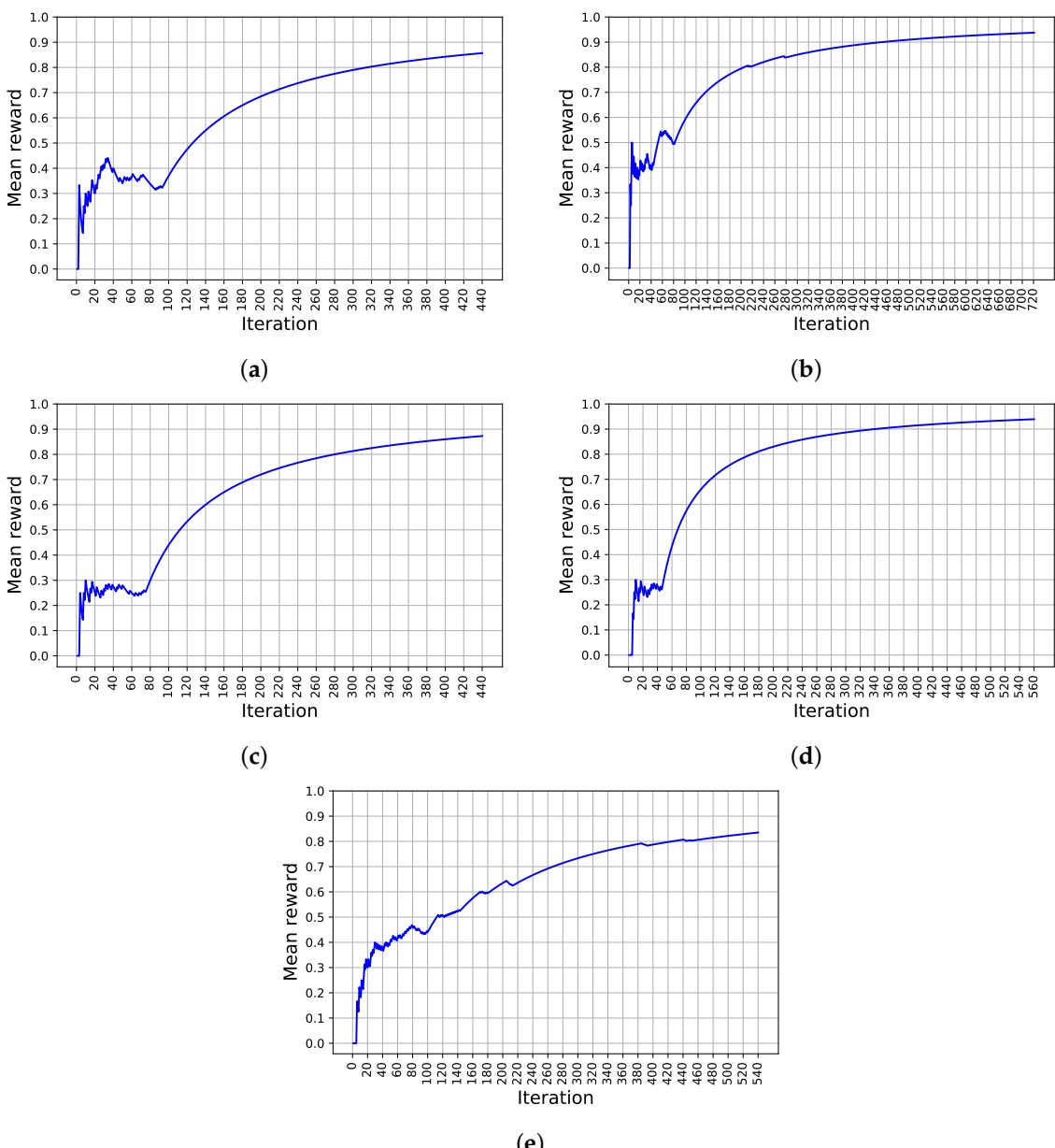

**Figure 8.** Mean reward trending curve captured at every iteration: (**a**) Peer 1 Org 1; (**b**) CouchDB for Peer 1 Org 1; (**c**) Peer 1 Org 2; (**d**) D CouchDB for Peer 1 Org 2; (**e**) Orderer.

## 5. Discussion

In this section, we present a discussion regarding the advantages of the proposed approach, the envisioned application of the model, and any possible future work.

Advantage: The special reinforcement learning algorithm, multi-armed contextual bandit accelerates the learning process. From our results, we can see that it takes less than 1000 iterations to converge the mean reward curve in every case. As the technique is exploratory, it can introduce completely new solutions that were never even observed. Moreover, if there exists any bias during the data annotation procedure, supervised learning will learn from the inherited bias. In this sense, reinforcement techniques are preferable.

Although scalability is a well-known problem in blockchain, it is mostly applicable to public blockchain systems. Private blockchain systems are designed to be scalable [16,44]. Even so, it might be increasingly difficult with the addition of RL techniques. To mitigate this, the training for RL techniques must be repeated on a regular basis in the backend.

The proposed method uses multi-armed bandit learning which is less complex than other algorithms and can be adopted for both small-scale and large-scale networks.

Envisioned application: Next, we explore how this model can be utilised within a real system. One possibility is that it can be integrated with a blockchain system in a very similar way to how we have integrated the ML module with the job scheduler. However, the role of this envisioned system will be different from ours. For example, the main purpose of the ML module presented in this work is to prepare a model by training it appropriately. Once the model is ready, we can extend the ML module for making decisions based on current resource consumption and then extend the job scheduler to pass on this decision taken by the ML module to the blockchain platform. One way this can be achieved is by the creating additional data models of *decision action* as presented in Table 4. Then, we can extend the flow in Figure 5 with the flow presented in Figure 9.

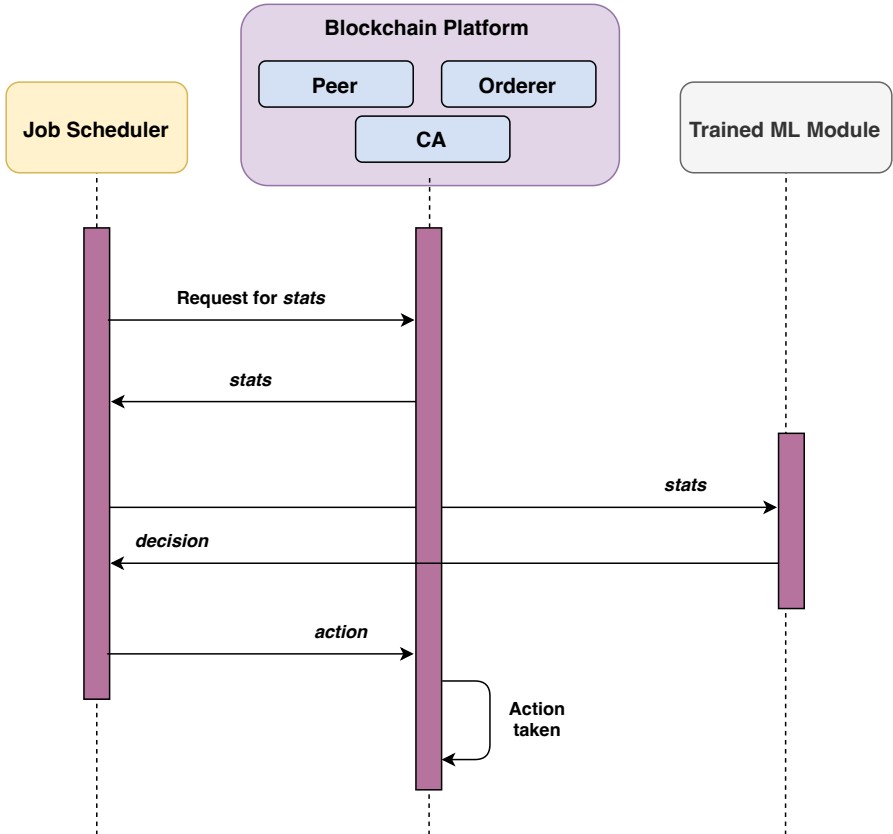

**Figure 9.** Envisioned protocol flow using the trained ML module.

**Table 4.** Additional data model.

| |
|---|
| *decision* ≜ ⟨*UP*, *DOWN*, *STABLE*⟩ |
| *action* ≜ ⟨*UP* ∨ *DOWN*⟩ |

To explain our vision, the *decision* represents one of the three decisions the ML module can take: *UP*, *DOWN*, and *STABLE*. Based on the value of the decision, the job scheduler will request one of the actions: *UP* or *DOWN*. For example, if the job scheduler receives an *UP* decision, it will send the *UP* value as part of the actions to the blockchain platform. The blockchain platform will consequently either scale up the resources of the corresponding entity or not.

Comparative Analysis: Next, we present a comparative analysis between existing works with our proposed work against a set of criteria where the selected criteria represent the crucial properties used in this work. The comparative analysis is presented in a tabular

format in Table 5 for better visualisation. In Table 5, the symbol '●' is used to denote that a certain criterion is fulfilled by the respective work, whereas '○' denotes that a certain criterion is not fulfilled. As evident from Table 5, our work satisfies all the criteria where the works from other researchers have failed to satisfy all the selected criteria.

**Table 5.** Comparative analysis of existing works with the proposed work.

| Research Work | Load Balancing | ML Approach | Container | Private Blockchain |
|---|:---:|:---:|:---:|:---:|
| Rossi et al. [9] | ○ | Reinforcement Learning | ● | ○ |
| Goli et al. [10] | ○ | Linear Regression, Random Forest, and Support Vector Regressor | ● | ○ |
| Dang-Quang et al. [11] | ● | Bidirectional LSTM | ● | ○ |
| Hamid et al. [13] | ● | Reinforcement Learning | ○ | ○ |
| Xinjie et al. [14] | ○ | ○ | ● | ○ |
| Our work | ● | Reinforcement Learning | ● | ● |

**Future Work:** In this work, our focus has been to present the motivation and methods for creating an ML model which will be suitable for load balancing the resources in the Hyperledger Fabric network. In future, we would like to integrate this model into the whole system as envisioned above and investigate its utility. In addition, we would also like to explore how this model can be integrated with other private blockchain platforms.

### 6. Concluding Remarks

The adoption of private blockchain platforms within different enterprise applications (including industrial control systems [45]) will be an important trend in the coming years. Hyperledger Fabric, being one of the most popular private blockchain platforms, will play a key role in this trend. However, load balancing of resourcing within the node of the blockchain is an important issue which is often overlooked. In this article, we have presented the first-ever reinforcement-learning-based machine learning (ML) model for load balancing of resources for Hyperledger Fabric. We have presented the architecture of the proposed method and analysed the interactions of different components of the architecture via protocol flows. We have detailed out the procedures for data collection and training an optimal ML model. We have elaborated how this model can be integrated within any Fabric application. Industrial applications might be required to serve a huge amount of input loads, and without such a load balancer, the corresponding node of the blockchain solution needs to be dynamically scaled up and down. Otherwise, the performance would degrade significantly. The approach presented in this article could be an effective tool in this situation; hence, this ML-based load balancing mechanism could be an important component to deploy blockchain solutions in industrial applications. In addition, the proposed solution could be applied to any industrial application and service business, including energy sectors, finance, healthcare, education, and smart industries [46,47]. It is to be noted that even though we have focused our proposal on Hyperledger Fabric, it can be easily integrated with other private blockchain platforms which rely on a container-based model. Thus, we strongly believe that the proposal will open up a new avenue of research within the private blockchain domain.

**Author Contributions:** Conceptualization, R.A., M.A. and M.S.F.; methodology, R.A., M.A. and M.S.F.; software, M.S.I.B. and R.S.R.; investigation, M.A. and M.S.F.; resources, R.A. and M.S.F.; writing—original draft, R.A., M.A., M.S.I.B., R.S.R. and M.S.F.; preparation, R.A., M.A. and M.S.F.; writing—review and editing, R.A. and M.S.F.; project administration, R.A.; funding acquisition, R.A. All authors have read and agreed to the published version of the manuscript.

**Funding:** The Deanship of Scientific Research (DSR) at King Abdulaziz University (KAU), Jeddah, Saudi Arabia has funded this project, under grant No. (G-195-612-1443).

**Data Availability Statement:** Not applicable.

**Conflicts of Interest:** The authors declare no conflict of interest.

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
