# Peer review of "A Reinforcement-Learning-Based Model for Resilient Load Balancing in Hyperledger Fabric"

_processes, doi:10.3390/pr10112390_

Round 1

Reviewer 1 Report

The authors proposed a reinforcement learning-based model for resilient load balancing in hyperledger fabric, which seems to be an appropriate approach. However, there are the following concerns that authors need to address, which is mentioned as follows:

1) In section 1, the authors discussed the existing works of the proposed system, which needs to be extended and improved to better understand the research gaps. The authors should present a comparative analysis of the existing works to further show the various research gaps.

2) The authors have performed resilient load balancing in hyperledger fabric using a private blockchain network, but they did not discuss any access control mechanism which can make the system inefficient and insecure.

3) The authors did not consider any consensus mechanism with the blockchain network which overall affects the security and privacy of the system.

4) The authors have not mentioned the scalability and reliability of the proposed system using a private blockchain network which can make it inefficient further deteriorating the performance.  

5) However, the private blockchain network considered in the proposed system seems to be a secure and preserved approach, but it can also suffer from various security attacks. For that, the authors need to perform a security analysis of the proposed system.

6) In subsection 4.3, the procedure of the obtained reward to evaluate the performance of the agent is not clear and appropriate. They should clearly mention the parameters and the mechanism for calculating the reward for the RL agent.

7) The authors have proposed a resilient load balancing in the hyperledger fabric. Still, there is no discussion on the incurred energy consumption which affects the resiliency and efficiency of the proposed system.

Author Response

Reply to Reviewer1’ Comments

We sincerely thank the editors to handle the paper and thank all the reviewers for providing valuable feedback to help us improve the quality of this paper. We have carefully considered all the review comments and revised the paper accordingly to address or clarify all the issues raised in the reviews. The following is a list of item-by-item responses to the review comments.

[Response to Reviewer 1’s Comments]

The authors proposed a reinforcement learning-based model for resilient load balancing in hyperledger fabric, which seems to be an appropriate approach. However, there are the following concerns that the authors need to address, which is mentioned as follows:

Comment 1: In section 1, the authors discussed the existing works of the proposed system, which needs to be extended and improved to better understand the research gaps. The authors should present a comparative analysis of the existing works to further show the various research gaps. 

Response: Thank you for this comment. The literature was closely reviewed and unfortunately, none of the existing works, to the best of our knowledge, presented a similar approach for any private blockchain system. In order to highlight this research gap, we have added a paragraph in Section 1 (Introduction section). Kindly check the highlighted texts in Section 1.

Furthermore, we have presented a comparative analysis with the relevant existing research works using a table where the existing relevant research works have been compared with our work against a set of criteria. This clearly highlights the research gap in the literature and how our work fills the gap. This comparative analysis can be found in a separate paragraph called “Comparative Analysis” in Section 5 (Discussion). Kindly check the highlighted texts in Section 5.  

Comment 2: The authors have performed resilient load balancing in hyperledger fabric using a private blockchain network, but they did not discuss any access control mechanism which can make the system inefficient and insecure. 

Response: We would like to thank the reviewer for this valid comment as it would be informative to discuss the access control mechanism. However, the scope of this paper is load balancing in private blockchains and the access control related to security is out of the scope of this research.

Comment 3: The authors did not consider any consensus mechanism with the blockchain network which overall affects the security and privacy of the system. 

Response: Thank you for the comment. Hyperledger Fabric currently supports a pluggable consensus algorithm meaning different types of CFT (Crush-fault Tolerant) and BFT (Byzantine Fault Tolerant) consensus algorithms can be retro-fitted. The current version of Hyperledger Fabric supports a RAFT based CFT consensus algorithm. We have used this algorithm in our experiments. We have added an explanatory line regarding this in the manuscript. Kindly check the highlighted texts in  Section 3.2.1.

Comment 4: The authors have not mentioned the scalability and reliability of the proposed system using a private blockchain network which can make it inefficient further deteriorating the performance. 

Response: Thank you for carefully reading the manuscript and raising this issue. We have added the following paragraph in the Discussion section. Kindly check the highlighted texts in lines  480-485 in the Discussion section.

“Although scalability is a well-known problem in blockchain, it is mostly applicable to public blockchain systems. Private blockchain systems are designed to be scalable [16,44]. Even so, it might be increasingly difficult with the addition of RL techniques. To mitigate this, the training for RL techniques must be repeated on a regular basis in the backend. The proposed method uses multi-armed bandit learning which is less complex than other algorithms and can be adopted for both small-scale and large-scale networks.”

Comment 5: However, the private blockchain network considered in the proposed system seems to be a secure and preserved approach, but it can also suffer from various security attacks. For that, the authors need to perform a security analysis of the proposed system. 

Response: We would like to thank the reviewer for this comment. Indeed, security is an important issue in the blockchain setting. However, the scope of this paper is load balancing in private blockchains and the security analysis is out of the scope of this research.

Comment 6:  In subsection 4.3, the procedure of the obtained reward to evaluate the performance of the agent is not clear and appropriate. They should clearly mention the parameters and the mechanism for calculating the reward for the RL agent.  

Response: Thank you for carefully reading the manuscript and raising this issue. We have updated the result section with additional explanations with respect to this comment. Kindly check the highlighted texts in Section 4.3.

Comment 7: The authors have proposed a resilient load balancing in the hyperledger fabric. Still, there is no discussion on the incurred energy consumption which affects the resiliency and efficiency of the proposed system. 

Response: We would like to thank the reviewer for their comment. The reviewer is indeed correct in saying that incurred energy consumption could affect the efficiency of a blockchain system. However, the proposed solution is for private blockchain systems which consume significantly less energy than any Proof-of-Work based (or similar consensus algorithm based) public blockchain systems. The proposed load balancing mechanism thus will consume less significant energy whose impact will be minimal in the respective private blockchain system.

Best regards,

The authors,

Reviewer 2 Report

This article discusses the first-ever solution to this significant problem by proposing an intelligent control system based on reinforcement learning for distributing the resources of Hyperledger Fabric. We present the architecture, discuss the protocol flows, outline the data collection methods, analyze the results and consider the potential applications of the proposed approach. The organization of the paper is good and the description is comprehensive. This study has a good application value.

We suggest describing the advantages and limitations of this architectural model. The description is to distinguish the proposed approach from previous research approaches.

additional comment:

1. Can this model be applied to other industrial fields? Provide a concise discussion supported by adequate references.
2. Provide a more comprehensive discussion of the differences between the proposed model and the development of previous similar models.
3. What are the requirements that fabric companies have to prepare in implementing this model? What added value does the company get? Tell
4. The stages of the research are clear and structured. However, please provide a brief review of the methodological differences in this study with the methodologies of previous studies.
5. Add an affirmation of the advantages of this model in the conclusion.

Author Response

Reply to Reviewers’ Comments

We sincerely thank the editors to handle the paper and thank all the reviewers for providing valuable feedback to help us improve the quality of this paper. We have carefully considered all the review comments and revised the paper accordingly to address or clarify all the issues raised in the reviews. The following is a list of item-by-item responses to the review comments.

Best regards,

The authors,

[Response to Reviewer 2’s Comments]

This article discusses the first-ever solution to this significant problem by proposing an intelligent control system based on reinforcement learning for distributing the resources of Hyperledger Fabric. We present the architecture, discuss the protocol flows, outline the data collection methods, analyze the results and consider the potential applications of the proposed approach. The organization of the paper is good and the description is comprehensive. This study has a good application value.

Comment 1: We suggest describing the advantages and limitations of this architectural model. The description is to distinguish the proposed approach from previous research approaches. 

Response: Thank you for your suggestion. In order to highlight this research gap, we have added a paragraph in Section 1 (Introduction section). Kindly check the highlighted texts in Section 1. Furthermore, we have presented a comparative analysis with the relevant existing research works using a table where the existing relevant research works have been compared with our work against a set of criteria. This clearly highlights the research gap in the literature and how your work fills the gap. This comparative analysis can be found in a separate paragraph called “Comparative Analysis” in Section 5 (Discussion). The newly added texts and the table have been highlighted in blue texts. 

Comment 2: 1. Can this model be applied to other industrial fields? Provide a concise discussion supported by adequate references. 

Response: Thank you for your comment. The proposed solution adopted a novel reinforcement learning which could be utilised for load balancing in Hyperledger Fabric. This could be applied to other industrial applications and service businesses including energy sectors, finance, healthcare, education and smart industries. We have added a few lines of explanatory texts with relevant references in Section 6. Kindly check the highlighted texts in Section 6.

Comment 3:  Provide a more comprehensive discussion of the differences between the proposed model and the development of previous similar models. 

Response: Thank you for this comment. The literature was closely reviewed and unfortunately, none of the existing works, to the best of our knowledge, presented a similar approach for any private blockchain system. In order to highlight this research gap, we have added a paragraph in Section 1 (Introduction section). Kindly check the highlighted texts in Section 1. 

Furthermore, we have presented a comparative analysis with the relevant existing research works using a table where the existing relevant research works have been compared with our work against a set of criteria. This clearly highlights the research gap in the literature and how your work fills the gap. This comparative analysis can be found in a separate paragraph called “Comparative Analysis” in Section 5 (Discussion). The newly added texts and the table have been highlighted in blue texts. 

Comment 4: What are the requirements that fabric companies have to prepare in implementing this model? What added value does the company get? Tell 

Response: Thank you for your comment. It is envisioned that the proposed system will be packaged within the Hyperledger Fabric architecture. Whenever any organisation within a Hyperledger Fabric blockchain system will deploy the packaged Hyperledger Fabric blockchain, the proposed system will get deployed automatically. To increase the performance of the ML module, the corresponding node can utilize GPUs. 

Comment 5: The stages of the research are clear and structured. However, please provide a brief review of the methodological differences in this study with the methodologies of previous studies. 

Response: Thank you for your comment. The proposed solution is adopted for private blockchain networks like Hyperledger Fabric. The methodology is similar to previous work in terms of the main steps, however, the main differences in the simulation, parameters, and analysis as follows: 

  • Building blockchain networks including DApp, Job Scheduler, and Contextual Multi-Armed Bandits Module that does not require static training data.
  • Data collection using simulation scenario: 5 users from two organizations
  • Feature extraction: we do not only consider a single time moment record, rather we inspect a fixed window of previous time moment records for understanding the current resource usage trend.
  • Analyzing results using mean reward metric with training six agents

Please note that none of the existing works have explored the possibility of utilizing blockchain based load balancing mechanisms for private blockchain systems. In such, this is a unique and novel work. Kindly check the “Comparative Analysis” in Section 5 (Discussion) to understand the difference between our work and previous works.

Comment 6: Add an affirmation of the advantages of this model in the conclusion. 

Response: Thank you for your suggestion. We have added a few lines of explanatory texts in Section 6. Kindly check the highlighted texts in Section 6.

Sincerely,

The Authors

Round 2

Reviewer 1 Report

The Paper is now in accepted form.